# Color-Tunable Indolizine-Based Fluorophores and Fluorescent pH Sensor

**DOI:** 10.3390/molecules27010012

**Published:** 2021-12-21

**Authors:** Taegwan Kim, Jonghoon Kim

**Affiliations:** 1Department of Chemistry, Soongsil University, Seoul 06978, Korea; lever19786@soongsil.ac.kr; 2Integrative Institute of Basic Science, Soongsil University, Seoul 06978, Korea

**Keywords:** indolizine, fluorescence, fluorophore design, pH sensor, intramolecular charge transfer (ICT)

## Abstract

A new fluorescent indolizine-based scaffold was developed using a straightforward synthetic scheme starting from a pyrrole ring. In this fluorescent system, an *N*,*N*-dimethylamino group in the aryl ring at the C-3 position of indolizine acted as an electron donor and played a crucial role in inducing a red shift in the emission wavelength based on the ICT process. Moreover, various electron-withdrawing groups, such as acetyl and aldehyde, were introduced at the C-7 position of indolizine, to tune and promote the red shift of the emission wavelength, resulting in a color range from blue to orange (462–580 nm). Furthermore, the ICT effect in indolizine fluorophores allowed the design and development of new fluorescent pH sensors of great potential in the field of fluorescence bioimaging and sensors.

## 1. Introduction

Optical fluorescence imaging for the non-invasive study of living systems has contributed to the organic fluorophore and fluorescent probe exploitation in the fields of life sciences, medicine, chemical biology, and biotechnology [1,2,3,4,5,6,7,8]. Due to a high demand for the development of novel fluorescent core skeletons, considerable progress has been made over the past few decades, which has resulted in the synthesis of various new fluorophores characterized by specific photophysical properties allowing the development of novel fluorescent probes [9,10,11,12,13]. As an example, indolizine-based organic molecules were successfully exploited in new fluorophores and fluorescent probes [14,15,16,17,18,19,20]. Synthetic methods for various types of indolizine scaffolds have led to the development of well-defined fluorophore scaffolds with an in-depth studied structure-fluorescence relationship. Figure 1a reports an emission-tunable and predictable indolizine-based fluorescent core skeleton, 9-aryl-1,2-dihydropyrrolo[3,4-*b*]indolizin-3-one, named Seoul-Fluor, and various fluorescent probes developed by Park et al. Perturbation of the electron density of the substituents at the C-7 and C-9 positions in Seoul-Fluor dramatically changed the emission wavelength, allowing a systematic tuning of the emission in the range 420–613 nm [21,22,23]. Moreover, computational studies provided guidelines for tuning the emission wavelength of Seoul-Fluor and developing Seoul-Fluor analogs with improved molar absorptivity [24]. More recently, Kim et al. prepared a new colorful fluorescent scaffold, named Kaleidolizine (KIz), with unique aggregation-induced emission (AIE) properties, on which the resulting fluorogenic probes were based [25,26]. However, previous studies typically focused on the synthesis of indolizine via the cycloaddition reaction of pyridinium ylide with alkenes or alkynes, with results critically influenced by the substrates [27,28,29]. Thus, most of these reactions are limited by incorporation of various functional groups at specific indolizine positions [30]. To address this, substantial efforts have been devoted to the development of straightforward synthetic methods for the incorporation of functional groups into indolizine. An indolizine-based full-color-tunable (405–616 nm) fluorophore, named C3-Indo-Fluoro, was developed by Lan et al. through the metal catalytic C–H activation of indolizine [31]. Despite the notable advances in indolizine-based fluorophores, the development of a straightforward synthetic strategy for the versatile functionalization of the indolizine core, providing convenient access to new indolizine fluorophores with unique photophysical properties, is challenging. To this end, we conceived a new synthetic route to prepare 3,7-disubstitued indolizine from the pyrrole ring (Figure 1b). The methodology reported in this study revealed a systematic tunability of the emission wavelength (462–580 nm) through the perturbation of the substituent electron density based on the intramolecular charge transfer (ICT) process [32]. Finally, the introduction of a pH-sensitive functional group into this new indolizine-based fluorescent scaffold allowed the successful development of a fluorescent pH sensor.

## 2. Results and Discussion

Based on the retrosynthetic analysis, the synthetic pathway employed to provide 3,7-disubstitued indolizine compounds is illustrated in Figure 1. First, the SN2 reaction of 1*H*-pyrrole-2-carboxaldehyde (**1**) with ethyl 4-bromobutyrate afforded compound **2**, followed by aldol condensation using sodium hydride (NaH), resulting in the cyclic compound **3**. Regioselective bromination of compound **3** with *N*-bromosuccinimide (NBS) generated compound **4**. Afterwards, the dehydrogenative oxidation with manganese dioxide (MnO_2_) generated the indolizine compound **5** with bromo and ester functional groups at the C-3 and C-7 positions, respectively, which allow facile conversion into other functional groups. Finally, to introduce an aryl moiety into the C-3 position for the new π-conjugated indolizine, we adopted the Suzuki cross-coupling reaction. The optimized cross-coupling conditions were conventional heating in water in the presence of palladium(II) acetate (Pd(OAc)_2_) and tetrabutylammonium bromide (TBAB) without any organic co-solvents [33]. Under the optimized reaction conditions, we successfully prepared a series of 7-aryl substituted indolizine-based fluorescent compounds with various R^1^ substituents to change the electronic characteristics of the phenyl ring.

Our systematic evaluation of the structure–photophysical property relationship included the maximum absorption and emission wavelengths, Stokes shifts, and quantum yields measurements (Table 1). Normalized absorption and emission spectra of each compound are shown in Appendix A in Appendix A. Interestingly, for all the functional groups introduced at the R^1^ position (from an electron-donating group, EDG, such as the methoxy group, to an electron-withdrawing group, EWG, such as the cyano group), the maximum absorption and emission wavelength change was negligible. However, compound **9** bearing an *N*,*N*-dimethylamino group, which is the most electron-donating group of the series, exhibited a red shift in the emission spectra (533 nm), compared to other compounds (462–492 nm). We hypothesized that this feature originated from the ICT process between the *N*,*N*-dimethylamino group at the R^1^ position and the ester group at the C-7 position of indolizine. Contrarily to other donating groups, with the ester group acting as an electron acceptor, in this electron-push-pull π-conjugated indolizine-based system, the *N*,*N*-dimethylamino group acted as a powerful electron donor, triggering an ICT process. In the electronic push-pull system in fluorescent scaffolds, the effects of the substituent electronic characteristics on the energy gap between the highest occupied molecular orbital (HOMO) and the lowest unoccupied molecular orbital (LUMO) resulted in a red or blue shift in emission [34]. In the case of compound **9**, the *N*,*N*-dimethylamino group on the phenyl ring might increase the HOMO energy level, with a minimal effect on the LUMO energy level, causing a red shift of the emission wavelength. Moreover, ICT causes a large change in the excited state dipole moment, promoting a solvent-dependent emission shift [35]. To determine the effect of ICT, we further studied the solvatochromic fluorescence emission change of compound **9**. As shown in Figure 2, the solvent polarity had limited effects on the absorption wavelength. Conversely, an increase in solvent polarity induced a bathochromic shift of the emission wavelength from 485 nm in non-polar diethyl ether to 533 nm in polar methanol. This positive emission solvatochromism suggested that the ICT process was the dominant mechanism for the red-shift emission of compound **9**.

Based on this observation, we designed and synthesized indolizine-based fluorescent compounds to tune the emission wavelength, which can cover a wide range of colors. As the *N*,*N*-dimethylamino group at the R^1^ position could successfully reduce the HOMO-LUMO energy gap by increasing the HOMO energy level, we hypothesized that the EWG group at the C-7 position influences the stabilization of the LUMO state, allowing a red-shift emission. As shown in Figure 2, to introduce a stronger electron-withdrawing group than the ester group at the C-7 position, compound **9** was subjected to a hydrolysis reaction in the presence of sodium hydroxide (NaOH), to afford compound **14**. Compound **15** was generated through an amide coupling reaction with *N*,*O*-dimethylhydroxylamine after the activation of carboxylic acids using carbonyldiimidazole (CDI). The *N*-methoxy-*N*-methyl amide group (Weinreb amide) was converted into aldehyde or ketone after treatment with lithium aluminum hydride (LiAlH_4_) or methylmagnesium bromide (CH_3_MgBr), to produce compounds **16** and **17**, respectively. Compounds **16** and **17** bearing aldehyde and acetyl groups, respectively, which are stronger electron-withdrawing groups than ester, showed a 47 nm and 45 nm red shift of emission wavelength, respectively, compared to compound **9** (Table 1). Nevertheless, the carboxylic acid group at the C-7 position induced a 28 nm blue shift in emission (Table 1). This was reasonably due to the reduction of the electron-withdrawing strength upon the introduction of the carboxylic acid group at the C-7 position. Figure 3 clearly shows a positive solvatochromism (from 499 to 574 nm) and a reduction in fluorescence intensity under polar conditions in compound **17**. These more considerable bathochromic shifts and changes in fluorescence intensity compared to compound **9** can be ascribed to the strengthening the ICT process through the introduction of more electron-withdrawing acetyl groups than the ester group at the C-7 position of the indolizine fluorophores. Overall, the synthesized new indolizine fluorophore compounds covered blue to orange-red color emission ranges (462–580 nm). The normalized emission spectra of representative compounds **12** (blue), **14** (green), **9** (yellow), and **16** (red-orange) are shown in Figure 4a. The colors, brought about irradiation at 365 nm, were visible to the naked eye, as shown in the fluorescence image (Figure 4b).

Because ICT processes are extensively used in the design of fluorescent probe [36,37,38,39], we hypothesized that the unique features of this indolizine-based fluorescent scaffold could lead to new applications for environmental changes as a fluorescent pH sensor. Therefore, compound **14** was selected as a potential pH sensor probe, owing to its pH-responsive functional group, namely the *N*,*N*-dimethylamino group, at the R^1^ position and carboxylic acid group at the C-7 position as well as considering high solubility in water solution. The protonation and deprotonation processes of these pH-responsive groups can induce changes in their electronic characteristics according to their pKa values, generating subsequent dramatic shifts in the emission wavelength. As an example, under acidic conditions, protonation of the dialkylamino group at the R^1^ position can promote the blue shift of the emission wavelength through the reduction of electron-donating characteristics. Nevertheless, under neutral conditions, the deprotonation of carboxylic acid can reduce the electron-donating character, causing a red-shifted emission. To demonstrate this, the emission response of compound **14** at different pH values (3.0–7.0) was examined. We observed a blue shift of the emission wavelength from 529 nm to 473 nm and an improvement of fluorescence intensity with the pH decreasing from 7.0 to 3.0 (Figure 5a,b). This result suggested that the protonation and deprotonation of the functional groups on indolizine fluorophores perturbed the electronic states, which could control the ICT process. In neutral conditions, the dimethylamino group could strengthen the ICT process compared to protonated dimethylamino group in acid conditions, leading to a red shift of the emission wavelength and reduction of emission intensity. As shown in Figure 5c, the emission intensity at 473 nm gradually decreased at higher pH values, revealing the potential of compound **14** as a fluorescent pH sensor. The inflection point of the sigmoid curve for compound **14** was pH 4.07. Besides compound **14**, the emission response of compounds **9**, **12**, **16**, and **17** at different pH values (3.0, 5.0, and 7.0) was examined. As expected, compound **9,** containing a pH-responsive element such as a dimethylamino group, clearly shows a blue shift of the emission wavelength from 522 nm to 476 nm and improvement of fluorescence intensity with the pH decreasing from 7.0 to 3.0, similar to compound **14**. In addition, compounds **16** and **17** exhibit a more dramatic reduction of emission intensity with the pH increasing from 3.0 to 7.0 compared to compound **9**, which is caused by strengthening the ICT process through the introduction of a more electron-withdrawing group (aldehyde and acetyl group) at the C-7 position, consistent with the findings in a previous solvatochromic study (Figure 6c,d). On the other hand, in the case of compound **12** containing cyano group at the R^1^ position, the emission wavelength and intensity cannot be obviously changed depending on the pH value, which can be ascribed to less intense ICT through the introduction of an electron-withdrawing group into the R^1^ position (Figure 6b). Thus, we clearly demonstrated that the ICT phenomenon in the indolizine fluorophore allowed the successful development of fluorescent pH sensors by introducing pH-responsive functional groups such as *N*,*N*-dimethylamino, revealing the potential of this new indolizine fluorophore for versatile fluorescent probes to explore biological systems and environmental changes.

## 3. Materials and Methods

All commercially available reagents were used without further purification unless noted otherwise. Dried solvents were passed through a solvent purification system equipped with activated alumina columns (glass contours). NMR analyses were carried out using a JEOL ECZ500/S1 spectrometer (500 MHz, Jeol, Tokyo, Japan). The chemical shifts of the proton (^1^H) and carbon (^13^C) NMR spectra were reported in parts per million (δ), with respect to the internal standard tetramethylsilane (TMS) or to the residual solvent peak (CDCl_3_, ^1^H: 7.26, ^13^C: 77.00; DMSO-*d_6_*, ^1^H: 2.49, ^13^C: 39.52; CD_2_Cl_2_, ^1^H: 5.32, ^13^C: 53.8). The multiplicities of the ^1^H NMR peaks are reported as follows: s (singlet), d (doublet), t (triplet), q (quartet), quintet, m (multiplet), dd (doublet of doublets), dt (doublet of triplets), td (triplet of doublets), and br s (broad singlet). Signals marked with an asterisk (*) correspond to the peaks assigned to the minor rotamer conformation. The coupling constants (*J*) were reported in Hz. High-resolution mass spectrometry (HRMS) spectra were obtained on a compact QToF spectrometer (Bruker, Billerica, MA, USA) using the electrospray ionization (ESI) method. Low-resolution mass spectrometry (LRMS) spectra were obtained on a compact mass spectrometer (Advion, Ithaca, NY, USA) using the atmospheric pressure chemical ionization (APCI) method. To monitor the progress of the reactions, analytical thin-layer chromatography (TLC) analysis was performed using glass plates pre-coated with silica gel (60 F254, Merck, Darmstadt, Germany), and the components were observed under UV light (254 and 365 nm). Flash column chromatography was performed using silica gel 60 (230–400 mesh, Merck, Darmstadt, Germany). Absorption spectra were recorded using a UV-3101PC spectrophotometer (Shimadzu, Kyoto, Japan) or V-770 spectrophotometer (Jasco, Easton, MD, USA). Fluorescence emission spectra were obtained with a RF-5301PC Fluorescence spectrophotometer (Shimadzu, Kyoto, Japan), and the absolute quantum yield was measured by QE-2000 (Otsuka Electronics, Osaka, Japan).

*Ethyl 4-(2-formyl-1H-pyrrol-1-yl)butanoate* (**2**). To a solution of 1*H*-pyrrole-2-carbaldehyde (4.33 g, 45.5 mmol) in acetonitrile (ACN; 230 mL), cesium carbonate (Cs_2_CO_3_; 29.6 g, 91.0 mmol) was added. The reaction mixture was stirred at room temperature (r.t.) for 30 min. Ethyl 4-bromobutyrate (7.8 mL, 54.6 mmol) was slowly added to the resultant solution at 0 °C, and the reaction mixture was stirred at r.t. for 18 h. The crude reaction mixture was partitioned into water and ethyl acetate (EtOAc). The aqueous layer was extracted twice with ethyl acetate (EtOAc). The combined organic layers were dried over anhydrous magnesium sulfate (MgSO_4_), filtered, and evaporated. Subsequent silica-gel flash column chromatography afforded ethyl 4-(2-formyl-1*H*-pyrrol-1-yl)butanoate (**2**) (9.66 g, 44.6 mmol, 98%, *R_f_* = 0.70 (EtOAc/hexane = 1:5) as a pale yellow oil. ^1^H NMR (500 MHz, CDCl_3_) δ (ppm): 9.52 (d, *J* = 1.0 Hz, 1H), 6.94–6.92 (m, 1H), 6.22 (dd, *J* = 4.0, 3.0 Hz, 1H), 4.37 (t, *J* = 7.0 Hz, 2H), 4.12 (q, *J* = 7.0 Hz, 2H), 2.27 (t, *J* = 7.5 Hz, 2H), 2.08 (quintet, *J* = 7.5 Hz, 2H), 1.25 (t, *J* = 7.0 Hz, 3H). ^13^C NMR (125 MHz, CDCl_3_) δ (ppm): 179.3, 172.8, 131.5, 131.3, 125.0, 109.7, 60.5, 47.9, 30.8, 26.4, 14.2. LRMS (APCI +): calcd for C_11_H_16_NO_3_^+^ [M + H]^+^ 210.1, found 210.3.

*Ethyl 5,6-dihydroindolizine-7-carboxylate* (**3**). To a solution of compound **2** (1.96 g, 9.35 mmol) in ethanol (EtOH; 500 mL), sodium hydride (NaH; 0.41 g of 60 % dispersion in paraffin liquid, 10.3 mmol) was added portion-wise at 0 °C. The reaction mixture was stirred at 70 °C for 18 h and then cooled to r.t. The resultant mixture was quenched with saturated ammonium chloride (NH_4_Cl, aq). EtOH was evaporated under reduced pressure, and the crude mixture was partitioned into water and EtOAc. The aqueous layer was extracted twice with EtOAc. The combined layer was dried over anhydrous MgSO_4_, filtered, and evaporated. Subsequent silica-gel flash column chromatography afforded ethyl 5,6-dihydroindolizine-7-carboxylate (**3**) (1.39 g, 7.27 mmol, 78%, *R_f_* = 0.48 (EtOAc/hexane = 1:5)) as a pale white oil. ^1^H NMR (500 MHz, CDCl_3_) δ (ppm): 7.47 (s, 1H), 6.73 (t, *J* = 1.5 Hz, 1H), 6.35 (dd, *J* = 3.5, 1.5 Hz, 1H), 6.21 (dd, *J* = 4.0, 3.0 Hz, 1H), 4.25 (q, *J* = 7.0 Hz, 2H), 4.01(t, *J* = 7.0 Hz, 2H), 2.80 (td, *J* = 7.5, 1.5 Hz, 2H), 1.33 (t, *J* = 8.5 Hz, 3H). ^13^C NMR (125 MHz, CDCl_3_) δ (ppm): 167.1, 128.2, 128.0, 123.9, 119.3, 111.8, 109.7, 60.4, 43.9, 23.6, 14.3. LRMS (APCI +): calcd for C_11_H_14_NO_2_^+^ [M + H]^+^ 192.1, found 192.3.

*Ethyl 3-bromo-5,6-dihydroindolizine-7-carboxylate* (**4**). To a solution of compound **3** (0.50 g, 2.63 mmol) in tetrahydrofuran (THF; 2.6 mL), *N*-bromosuccinimide (NBS; 0.47 g, 2.63 mmol) was added portion-wise at r.t. After the reaction was complete, as indicated by TLC, the reactant was extracted with EtOAc twice, filtered, and evaporated. Subsequent silica-gel flash column chromatography afforded ethyl 3-bromo-5,6-dihydroindolizine-7-carboxylate (**4**) (0.64 g, 2.37 mmol, 90%, *R_f_* = 0.43 (EtOAc/hexane = 1:5)) as a pale white oil. ^1^H NMR (500 MHz, CDCl_3_) δ (ppm): 7.37 (t, *J* = 1.0 Hz, 1H), 6.33 (d, *J* = 4.0 Hz, 1H), 6.22 (d, *J* = 3.5 Hz, 1H), 4.25 (q, *J* = 7.5 Hz, 2H), 3.97 (t, *J* = 7.5 Hz, 2H), 2.80 (td, *J* = 7.5, 1.5 Hz, 2H), 1.33 (t, *J* = 7.0 Hz, 3H). ^13^C NMR (125 MHz, CDCl_3_) δ (ppm): 166.9, 129.6, 127.2, 119.5, 112.6, 112.2, 106.7, 60.5, 42.5, 23.2, 14.4. LRMS (APCI +): calcd for C_11_H_13_BrNO_2_^+^ [M + H]^+^ 270.0, found 270.2.

*Ethyl 3-bromoindolizine-7-carboxylate* (**5**). To a solution of compound **4** (0.71 g, 3.0 mmol) in dichloromethane (DCM; 150 mL) manganese dioxide (MnO_2_; 13.0 g, 150 mmol) was added. The reaction mixture was stirred under reflux for 18 h and then cooled to r.t. The mixture was filtered through Celite and solvent was evaporated. Subsequent silica-gel flash column chromatography afforded ethyl 3-bromoindolizine-7-carboxylate (**5**) (0.59 g, 2.20 mmol, 73%, *R_f_* = 0.46 (EtOAc/hexane = 1:6)) as a pale-yellow solid. ^1^H NMR (500 MHz, CDCl_3_) δ (ppm): 8.17 (s, 1H), 7.94 (d, *J* = 7.5 Hz, 1H), 7.21 (dd, *J* = 7.5, 2.0 Hz, 1H), 6.90 (d, *J* = 4.5 Hz, 1H), 6.77 (d, *J* = 4.0 Hz, 1H), 4.38 (q, *J* = 7.5 Hz, 2H), 1.41 (t, *J* = 7.5 Hz, 3H). ^13^C NMR (125 MHz, CDCl_3_) δ (ppm): 165.8, 132.4, 122.7, 122.1, 118.7, 117.5, 110.4, 105.3, 96.1, 61.0, 14.4. LRMS (APCI +): calcd for C_11_H_11_BrNO_2_^+^ [M + H]^+^ 268.0, found 268.2.

*Ethyl 3-phenylindolizine-7-carboxylate* (**6**). To a solution of compound **5** (0.27 g, 1.0 mmol) in water (3 mL), phenyl boronic acid (0.13 g, 1.1 mmol), potassium carbonate (K_2_CO_3_; 0.35 g, 2.5 mmol), tetrabutylammonium bromide (TBAB; 0.32 g, 1.0 mmol), and palladium(II) diacetate (Pd(OAc)_2_; 4 mg, 0.02 mmol) were added. The reaction mixture was stirred at 70 °C for 2 h. The crude mixture was extracted twice with EtOAc. The combined organic layers were dried over anhydrous MgSO_4_, filtered, and evaporated. Subsequent silica-gel flash column chromatography afforded ethyl 3-phenylindolizine-7-carboxylate (**6**) (0.22 g, 0.82 mmol, 82%, *R_f_* = 0.43 (EtOAc/hexane = 1:6)) as a yellow oil. ^1^H NMR (500 MHz, CDCl_3_) δ (ppm): 8.24 (s, 1H), 8.23 (dt, *J* = 7.5, 0.5 Hz, 1H), 7.58 (dd, *J* = 8.5, 1.5 Hz, 2H), 7.50 (t, *J* = 7.5 Hz, 2H), 7.39 (tt, *J* = 7.0, 1.5 Hz, 1H), 7.07 (dd, *J* = 7.5, 1.5 Hz, 1H), 6.96 (d, *J* = 4.0 Hz, 1H), 6.83 (d, *J* = 4.5 Hz, 1H), 4.38 (q, *J* = 7.5 Hz, 2H), 1.41 (t, *J* = 7.0 Hz, 3H). ^13^C NMR (125 MHz, CD_2_Cl_2_) δ (ppm): 166.1, 133.0, 132.1, 129.4, 128.5, 128.4, 128.1, 123.5, 121.8, 118.9, 116.2, 110.0, 105.6, 61.1, 14.5. HRMS (ESI +): calcd for C_17_H_15_NO_2_Na^+^ [M + Na]^+^ 288.1000, found 288.0995.

*Ethyl 3-(p-tolyl)indolizine-7-carboxylate* (**7**). To a solution of compound **5** (0.27 g, 1.0 mmol) in water (3 mL), p-tolylboronic acid (0.15 g, 1.1 mmol), K_2_CO_3_ (0.35 g, 2.5 mmol), TBAB (0.32 g, 1.0 mmol), and Pd(OAc)_2_ (4 mg, 0.02 mmol) were added. The reaction mixture was stirred at 70 °C for 2 h. The crude mixture was extracted twice with EtOAc. The combined organic layers were dried over anhydrous MgSO_4_, filtered, and evaporated. Subsequent silica-gel flash column chromatography afforded ethyl 3-(*p*-tolyl)indolizine-7-carboxylate (**7**) (0.27g, 0.95 mmol, 95%, *R_f_* = 0.43 (EtOAc/hexane = 1:6)) as a yellow oil. ^1^H NMR (500 MHz, CDCl_3_) δ (ppm): 8.24 (s, 1H), 8.19 (dt, *J* = 7.5, 1.0 Hz, 1H), 7.47 (d, *J* = 8.0 Hz, 2H), 7.31 (d, *J* = 8.0 Hz, 2H), 7.05 (dd, *J* = 7.5, 2.0 Hz, 1H), 6.93 (d, *J* = 4.0 Hz, 1H), 6.81 (d, *J* = 3.5 Hz, 1H), 4.38 (q, *J* = 7.0 Hz, 2H), 2.43 (s, 3H), 1.41 (t, *J* = 7.0 Hz, 3H). ^13^C NMR (125 MHz, CDCl_3_) δ (ppm): 166.1, 137.7, 132.3, 129.7, 128.8, 128.2, 128.1, 123.5, 121.4, 118.2, 115.7, 109.7, 105.2, 60.8, 21.3, 14.4. HRMS (ESI +): calcd for C_18_H_17_NO_2_Na^+^ [M + Na]^+^ 302.1157, found 302.1151.

*Ethyl 3-(4-methoxyphenyl)indolizine-7-carboxylate* (**8**). To a solution of compound **5** (0.27 g, 1.0 mmol) in water (3 mL), (4-methoxyphenyl)boronic acid (0.17 g, 1.1 mmol), K_2_CO_3_ (0.35 g, 2.5 mmol), TBAB (0.32 g, 1.0 mmol), and Pd(OAc)_2_ (4 mg, 0.02 mmol) were added. The reaction mixture was stirred at 70 °C for 2 h. The crude mixture was extracted twice with EtOAc. The combined organic layers were dried over anhydrous MgSO_4_, filtered, and evaporated. Subsequent silica-gel flash column chromatography afforded ethyl 3-(4-methoxyphenyl)indolizine-7-carboxylate (**8**) (0.25 g, 0.83 mmol, 83%, *R_f_* = 0.25 (EtOAc/hexane = 1:6)) as a yellow solid. ^1^H NMR (500 MHz, CDCl_3_) δ (ppm): 8.23 (s, 1H), 8.14 (d, *J* = 7.5 Hz, 1H), 7.49 (d, *J* = 8.5 Hz, 2H), 7.06–7.03 (m, 3H), 6.89 (d, *J* = 4.5 Hz, 1H), 6.81 (d, *J* = 4.0 Hz, 1H), 4.37 (q, *J* = 7.0 Hz, 2H), 3.88 (s, 3H), 1.41 (t, *J* = 7.5 Hz, 3H). ^13^C NMR (125 MHz, CDCl_3_) δ (ppm): 166.1, 159.2, 132.1, 129.6, 128.0, 124.1, 123.4, 121.2, 118.0, 115.5, 114.5, 109.6, 105.1, 60.8, 55.4, 14.4. HRMS (ESI +): calcd for C_18_H_17_NO_3_Na^+^ [M + Na]^+^ 318.1106, found 318.1101.

*Ethyl 3-(4-(dimethylamino)phenyl)indolizine-7-carboxylate* (**9**). To a solution of compound **5** (0.27 g, 1.0 mmol) in water (3 mL), (4-(dimethylamino)phenyl)boronic acid (0.18 g, 1.1 mmol), K_2_CO_3_ (0.35 g, 2.5 mmol), TBAB (0.32 g, 1.0 mmol), and Pd(OAc)_2_ (4 mg, 0.02 mmol) were added. The reaction mixture was stirred at 70 °C for 2 h. The crude mixture was extracted twice with EtOAc. The combined organic layers were dried over anhydrous MgSO_4_, filtered, and evaporated. Subsequent silica-gel flash column chromatography afforded ethyl 3-(4-(dimethylamino)phenyl)indolizine-7-carboxylate (**9**) (0.24 g, 0.77 mmol, 77%, *R_f_* = 0.25 (EtOAc/hexane = 1:6)) as a yellow solid. ^1^H NMR (500 MHz, CDCl_3_) δ (ppm): 8.22 (s, 1H), 8.17 (d, *J* = 7.5 Hz, 1H), 7.44 (d, *J* = 9.0 Hz, 2H), 7.02 (dd, *J* = 7.5, 2.0 Hz, 1H), 6.87 (d, *J* = 4.0 Hz, 1H), 6.84 (d, *J* = 9.0 Hz, 2H), 6.80 (d, *J* = 4.0 Hz, 1H), 4.37 (q, *J* = 7.5 Hz, 2H), 3.03 (s, 6H), 1.41 (t, *J* = 7.5 Hz, 3H). ^13^C NMR (125 MHz, CDCl_3_) δ (ppm): 166.2, 149.9, 131.8, 129.1, 128.8, 123.4, 121.3, 119.3, 117.4, 115.1, 112.5, 109.3, 105.1, 60.7, 40.4, 14.4. HRMS (ESI +): calcd for C_19_H_21_N_2_O_2_^+^ [M + H]^+^ 309.1603, found 309.1598.

*Ethyl 3-(4-fluorophenyl)indolizine-7-carboxylate* (**10**). To a solution of compound **5** (0.27 g, 1.0 mmol) in water (3 mL), (4-fluorophenyl)boronic acid (0.15 g, 1.1 mmol), K_2_CO_3_ (0.35 g, 2.5 mmol), TBAB (0.32 g, 1.0 mmol), and Pd(OAc)_2_ (4 mg, 0.02 mmol) were added. The reaction mixture was stirred at 70 °C for 2 h. The crude mixture was extracted twice with EtOAc. The combined organic layers were dried over anhydrous MgSO_4_, filtered, and evaporated. Subsequent silica-gel flash column chromatography afforded ethyl 3-(4-fluorophenyl)indolizine-7-carboxylate (**10**) (0.24 g, 0.84 mmol, 84%, *R_f_* = 0.43 (EtOAc/hexane = 1:6)) as a yellow solid. ^1^H NMR (500 MHz, CDCl_3_) δ (ppm): 8.24 (s, 1H), 8.12 (d, *J* = 7.5 Hz, 1H), 7.55–7.51 (m, 2H), 7.20 (t, *J* = 8.5 Hz, 2H), 7.07 (dd, *J* = 7.5, 2.0 Hz, 1H), 6.91 (d, *J* = 4.0 Hz, 1H), 6.81 (d, *J* = 4.0 Hz, 1H), 4.38 (q, *J* = 7.0 Hz, 2H), 1.41 (t, *J* = 7.0 Hz, 3H). ^13^C NMR (125 MHz, CDCl_3_) δ (ppm): 165.9, 163.2, 161.2, 132.4, 130.0, 129.9, 127.8, 127.0, 123.5, 121.1, 118.5, 116.2, 116.0, 115.9, 109.9, 105.2, 60.8, 14.4. LRMS(ESI+): calcd for C_17_H_15_FNO_2_^+^ [M+H]^+^ 284.1, found 284.1. HRMS (ESI +): calcd for C_17_H_14_FNO_2_Na^+^ [M + Na]^+^ 306.0906, found 306.0901.

*Ethyl 3-(4-formylphenyl)indolizine-7-carboxylate* (**11**). To a solution of compound **5** (0.27 g, 1.0 mmol) in water (3 mL), (4-formylphenyl)boronic acid (0.16 g, 1.1 mmol), K_2_CO_3_ (0.35 g, 2.5 mmol), TBAB (0.32 g, 1.0 mmol), and Pd(OAc)_2_ (4 mg, 0.02 mmol) were added. The reaction mixture was stirred at 70 °C for 90 min. The crude mixture was extracted twice with EtOAc. The combined organic layers were dried over anhydrous MgSO_4_, filtered, and evaporated. Subsequent silica-gel flash column chromatography afforded ethyl 3-(4-formylphenyl)indolizine-7-carboxylate (**11**) (0.19 g, 0.63 mmol, 63%, *R_f_* = 0.11 (EtOAc/hexane = 1:6)) as a yellow solid. ^1^H NMR (500 MHz, CDCl_3_) δ (ppm): 10.05 (s, 1H), 8.31 (d, *J* = 7.5 Hz, 1H), 8.26 (s, 1H), 8.00 (d, *J* = 6.5 Hz, 2H), 7.76 (d, *J* = 8.0 Hz, 2H), 7.15 (dd, *J* = 7.5, 1.5 Hz, 1H), 7.07 (d, *J* = 4.0 Hz, 1H), 6.85 (d, *J* = 4.0 Hz, 1H), 4.39 (q, *J* = 7.5 Hz, 2H), 1.41 (t, *J* = 6.5 Hz, 3H). ^13^C NMR (125 MHz, CDCl_3_) δ (ppm): 191.4, 165.7, 137.6, 134.9, 133.8, 130.6, 127.7, 126.7, 123.5, 121.5, 119.6, 117.2, 110.6, 106.1, 61.0, 14.4. HRMS (ESI +): calcd for C_18_H_15_NO_3_Na^+^ [M + Na]^+^ 316.0950, found 316.0944.

*Ethyl 3-(4-cyanophenyl)indolizine-7-carboxylate* (**12**). To a solution of compound **5** (0.27 g, 1.0 mmol) in water (3 mL), (4-cyanophenyl)boronic acid (0.16 g, 1.1 mmol), K_2_CO_3_ (0.35 g, 2.5 mmol), TBAB (0.32 g, 1.0 mmol), and Pd(OAc)_2_ (4 mg, 0.02 mmol) were added. The reaction mixture was stirred at 70 °C for 2 h. The crude mixture was extracted twice with EtOAc. The combined organic layers were dried over anhydrous MgSO_4_, filtered, and evaporated. Subsequent silica-gel flash column chromatography afforded ethyl 3-(4-cyanophenyl)indolizine-7-carboxylate (**12**) (0.19 g, 0.66 mmol, 66%, *R_f_* = 0.14 (EtOAc/hexane = 1:6)) as a yellow solid. ^1^H NMR (500 MHz, CDCl_3_) δ (ppm): 8.26–8.25 (m, 2H), 7.77 (d, *J* = 8.5 Hz, 2H), 7.70 (d, *J* = 8.5 Hz, 2H), 7.15 (dd, *J* = 7.5, 1.5 Hz, 1H), 7.03 (d, *J* = 4.5 Hz, 1H), 6.85 (d, *J* = 4.5 Hz, 1H), 4.39 (q, *J* = 7.0 Hz, 2H), 1.41 (t, *J* = 7.0 Hz, 3H). ^13^C NMR (125 MHz, CDCl_3_) δ (ppm): 165.6, 136.2, 133.8, 132.9, 127.8, 126.0, 123.5, 121.3, 119.8, 118.7, 117.1, 110.7, 110.6, 106.1, 61.0, 14.3. HRMS (ESI +): calcd for C_18_H_14_N_2_O_2_Na^+^ [M + Na]^+^ 313.0953, found 313.0947.

*Ethyl 3-(3-cyanophenyl)indolizine-7-carboxylate* (**13**). To a solution of compound **5** (0.27 g, 1.0 mmol) in water (3 mL), (3-cyanophenyl)boronic acid (0.16 g, 1.1 mmol), K_2_CO_3_ (0.35 g, 2.5 mmol), TBAB (0.32 g, 1.0 mmol), and Pd(OAc)_2_ (4 mg, 0.02 mmol) were added. The reaction mixture was stirred at 70 °C for 2 h. The crude mixture was extracted twice with EtOAc. The combined organic layers were dried over anhydrous MgSO_4_, filtered, and evaporated. Subsequent silica-gel flash column chromatography afforded ethyl 3-(3-cyanophenyl)indolizine-7-carboxylate (**13**) (0.16 g, 0.54 mmol, 54%, *R_f_* = 0.14 (EtOAc/hexane = 1:6)) as a yellow solid. ^1^H NMR (500 MHz, CDCl_3_) δ (ppm): 8.26 (s, 1H), 8.17 (d, *J* = 7.0 Hz, 1H), 7.86 (t, *J* = 1.5 Hz, 1H), 7.82 (dt, *J* = 7.5, 1.5 Hz, 1H), 7.65 (dt, *J* = 7.5, 1.5 Hz, 1H), 7.61 (t, *J* = 8.0 Hz, 1H), 7.14 (dd, *J* = 7.5, 1.5 Hz, 1H), 6.99 (d, *J* = 4.0 Hz, 1H), 6.84 (d, *J* = 4.0 Hz, 1H), 4.39 (q, *J* = 7.0 Hz, 2H), 1.41 (t, *J* = 7.0 Hz, 3H). ^13^C NMR (125 MHz, CDCl_3_) δ (ppm): 165.7, 133.3, 133.1, 131.9, 131.3, 130.8, 130.0, 125.4, 123.5, 120.9, 119.4, 118.4, 116.6, 113.5, 110.6, 105.7, 61.0, 14.4. HRMS (ESI +): calcd for C_18_H_14_N_2_O_2_Na^+^ [M + Na]^+^ 313.0953, found 313.0947.

*3-(4-(Dimethylamino)phenyl)indolizine-7-carboxylic acid* (**14**). To a solution of compound **9** (0.37 g, 1.2 mmol) in EtOH (4 mL) and water (2 mL), sodium hydroxide (NaOH; 0.10 g, 2.4 mmol) was added. The reaction mixture was stirred at 60 °C. After the reaction was complete, as indicated by TLC, the mixture was acidified with 1 N hydrogen chloride (HCl; aq) to give a yellow precipitate. The resulting precipitate was filtered and dried to afford desired product, 3-(4-dimethylamino)phenyl)indolizine-7-carboxylic acid (**14**) (0.29 g, 1.0 mmol, 84%, *R_f_* = 0.20 (EtOAc/hexane = 2:1)) as a yellow solid. ^1^H NMR (500 Mz, DMSO-*d*_6_) δ (ppm): 12.67 (br s, 1H), 8.25 (d, *J* = 7.5 Hz, 1H), 8.15 (s, 1H), 7.43 (d, *J* = 9.0 Hz, 2H), 6.96 (dd, *J* = 7.5, 1.5 Hz, 1H), 6.92 (d, *J* = 4.0 Hz, 1H), 6.86–6.84 (m, 3H), 2.96 (s, 6H). ^13^C NMR (125 MHz, CDCl_3_)) δ (ppm): 171.5, 150.1, 131.8, 129.6, 129.2, 124.9, 121.4, 119.2, 116.1, 115.5, 112.6, 109.4, 106.2, 40.4. HRMS (ESI +): calcd for C_17_H_17_N_2_O_2_^+^ [M + H]^+^ 281.1290, found 281.1285.

*3-(4-(Dimethylamino)phenyl)-N-methoxy-N-methylindolizine-7-carboxamide* (**15**). To a solution of compound **14** (0.36 g, 1.3 mmol) in THF (13 mL), 1,1′-carbonyldiimidazole (CDI; 0.25 g, 1.6 mmol) was added at 0 °C. The reaction mixture was stirred at r.t for 30 min. To the resultant mixture, *N*,*O*-dimethylhydroxylamine hydrochloride (0.15 g, 1.5 mmol) was added at r.t and stirred for 18 h. The reaction mixture was partitioned into 1 N sodium hydroxide (NaOH; aq) solution and EtOAc. The aqueous solution was extracted twice with EtOAc. The combined organic solutions were washed with 1N NaOH, dried anhydrous MgSO_4_, filtered, and evaporated. Subsequent silica-gel flash column chromatography afforded 3-(4-(dimethylamino)phenyl)-*N*-methoxy-*N*-methylindolizine-7-carboxamide (**15**) (0.24 g, 0.75 mmol, 59%, *R_f_* = 0.22 (EtOAc/hexane = 1:1)) as an orange oil. ^1^H NMR (500 MHz, CDCl_3_) δ (ppm): 8.16 (d, *J* = 7.5 Hz, 1H), 8.02 (s, 1H), 7.44 (d, *J* = 9.0 Hz, 2H), 6.89 (dd, *J* = 7.5, 1.5 Hz, 1H), 6.86–6.83 (m, 3H), 6.72 (d, *J* = 4.0, 1H), 3.64/3.62* (s, 3H), 3.39/3.38* (s, 3H), 3.05*/3.03 (s, 6H). LRMS (APCI +): calcd for C_19_H_22_N_3_O_2_^+^ [M + H]^+^ 324.2, found 324.2.

*3-(4-(Dimethylamino)phenyl)indolizine-7-carbaldehyde* (**16**). To a solution of compound **15** (0.11 g, 0.35 mmol) in dry THF (3.5 mL), under an argon atmosphere, a solution of lithium aluminium hydride in THF (LiAlH_4_; 0.2 mL, 2.0 M) was added dropwise at 0 °C. The reaction mixture was stirred at r.t for 2 h. After the reaction was complete, as indicated by TLC, the mixture was quenched with saturated NH_4_Cl (aq), and then extracted twice with EtOAc. The combined organic layers were dried over anhydrous MgSO_4_, filtered, and evaporated. Subsequent silica-gel flash column chromatography afforded 3-(4-(dimethylamino)phenyl)indolizine-7-carbaldehyde (**16**) (47 mg, 0.18 mmol, 51%, *R_f_* = 0.32 (EtOAc/hexane = 1:5)) as a yellow solid. ^1^H NMR (500 MHz, CDCl_3_) δ (ppm): 9.76 (s, 1H), 8.18 (d, *J* = 7.5 Hz, 1H), 7.90 (s, 1H), 7.44 (d, *J* = 9.0 Hz, 2H), 6.98 (dd, *J* = 7.0, 1.5 Hz, 1H), 6.94 (d, *J* = 4.0 Hz, 1H), 6.91 (d, *J* = 4.5 Hz, 1H), 6.84 (d, *J* = 9.0 Hz, 2H), 3.04 (s, 6H). ^13^C NMR (125 MHz, CDCl_3_) δ (ppm): 189.4, 150.2, 131.6, 130.8, 129.2, 128.7, 125.5, 122.3, 118.7, 115.8, 112.3, 107.7, 106.3, 40.3. HRMS (ESI +): calcd for C_17_H_17_N_2_O^+^ [M + H]^+^ 265.1341, found 265.1335.

*1-(3-(4-(Dimethylamino)phenyl)indolizin-7-yl)ethan-1-one* (**17**). To a solution of compound **15** (0.11 g, 0.35 mmol) in dry THF (3.5 mL), under an argon atmosphere, a solution of methylmagnesium bromide in diethyl ether (CH_3_MgBr; 0.35 mL, 3.0 M) was added dropwise at 0 °C. The reaction mixture was stirred at 0 °C for 2 h. After the reaction was complete, as indicated by TLC, the mixture was quenched with saturated NH_4_Cl (aq), and then extracted twice with EtOAc. The combined organic layers were dried over anhydrous MgSO_4_, filtered, and evaporated. Subsequent silica-gel flash column chromatography afforded 1-(3-(4-(dimethylamino)phenyl)indolizin-7-yl)ethan-1-one (**17**) (78 mg, 0.28 mmol, 81%, *R_f_* = 0.26 (EtOAc/hexane = 1:3)) as a yellow solid. ^1^H NMR (500 MHz, CDCl_3_) δ (ppm): 8.16 (d, *J* = 8.0 Hz, 1H), 8.08 (d, *J* = 1.0 Hz, 1H), 7.44 (d, *J* = 8.5 Hz, 2H), 7.07 (dd, *J* = 7.0, 1.5 Hz, 1H), 6.88 (d, *J* = 4.0 Hz, 1H), 6.85 (d, *J* = 5.5 Hz, 2H), 6.83 (s, 1H), 3.03 (s, 6H), 2.58 (s, 3H). ^13^C NMR (125 MHz, CDCl_3_) δ (ppm): 195.6, 150.1, 131.4, 129.6, 129.2, 125.2, 123.4, 121.6, 119.1, 115.4, 112.5, 108.1, 106.4, 40.4, 25.6. HRMS (ESI +): calcd for C_18_H_19_N_2_O^+^ [M + H]^+^ 279.1497, found 279.1492.

## 4. Conclusions

In summary, a novel 3,7-disubstituted indolizine-based fluorophore, whose emission wavelengths were tunable to cover a wide color range from blue to red-orange (462–580 nm), was designed and synthesized through a straightforward synthetic scheme starting from the pyrrole ring. Most of the functional groups introduced into the aryl ring at the C-3 position of indolizine did not induce changes in the photophysical properties, such as the absorption and emission wavelength. However, compound **9** bearing an *N*,*N*-dimethylamino group at the aryl ring exhibited a pronounced red-shift emission spectra (533 nm), compared to the other compounds (462–492 nm). This result was caused by the ICT process between the *N*,*N*-dimethylamino group and the ester group at the C-7 position in the indolizine scaffold. Furthermore, indolizine-based fluorophores showed more evident red-shift emission spectra (580 nm) through strengthening the ICT process by the introduction of more electron-withdrawing groups than the ester group, finally successfully leading to the potential fluorescent pH probe. Unfortunately, these newly developed indolizine fluorophores show broad emission spectra, which can cause fluorescent cross-talk problems [40,41,42]. In addition, they have absorption wavelengths in the UV and blue light range (~400 nm). UV and blue light range possess higher energy than visible light (> 400 nm), can be more damaging to the live cell, and usually cause high scattering more than other color light, which occasionally leads to a lower bioavailability [43]. Despite these drawbacks, many other advantages, such as easy access for absorption and emission tunability and predictability guided by computational studies, have rendered indolizine-based fluorophores versatile tools to generate fluorescent and fluorogenic probes. The past decade has witnessed remarkable advances in the development of new indolizine-based fluorophores. Thus, we are sure that our indolizine-based fluorophores could act as the new molecular template for the development of fluorophores with better photophysical properties and fluorescent probes, which leads to providing convenience for live-cell imaging. The following study will be reported in due course.

## Data Availability

The data presented in this study are available in supplementary materials.

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
