# Peer review of "Color-Tunable Indolizine-Based Fluorophores and Fluorescent pH Sensor"

_molecules, 2021, doi:10.3390/molecules27010012_

Round 1
Reviewer 1 Report
The manuscript by Kim and Kim describes some spectral properties of new fluorophores on basis of indolizine. The development of new fluorophores is interesting topic. However, I have some remarks.
- The manuscript can be more interesting for potential readers, if authors will add discussion about practical application of new fluorophores. It can be noted that the emission spectra of fluorophores are wide (broad bands in visible light). This property can limit the using of fluorophores. Additionally, the absorption in UV and blue light range is nonoptimal because high scattering in solutions. The comparison of proposed fluorophores with commercial analogues can improve manuscript.
- The pH responses of compounds 9, 17, 12 and 16 should be showed. What was the pH influence on parameters of emission spectra (maximal intensity at peak, width of spectra, and others)?
Reviewer 2 Report
Fluorescent dyes and probes are useful tools for in situ and real-time imaging of live biological systems. These authors developed a new fluorescent indolizine-based scaffold and synthesized a series of ICT-based dyes with tunable photophysical properties. In addition, the ICT effect in indolizine fluorophores allowed the design and development of new fluorescent pH sensors for possible bioimaging. A limitation is that the authors did not validate the applications of the probes in biological imaging. Overall, this work is interesting and may be published.
- Spectra changes of the probe at different pH values may be useful for readers.
- New compounds may be characterized by HRMS or elemental analysis.
- Beside the amino group, the phenolic hydroxyl group is also an efficient ICT donor, but this work did not contain it.
- Other ICT-based systems such as NBD probes (Chem. Soc. Rev., 2021, 50, 7436) may be cited for comparison with the indolizine-based scaffold.
Reviewer 3 Report
See attached file.

Round 2
Reviewer 1 Report
The manuscript has been improved. I have not other questions.
Reviewer 3 Report
The authors answered all my concerns completely.
I believe that now the study should be published as is.